# 'Unmasking Pain' through creativity: A phenomenological study of person-centred pain exploration for people living with chronic pain

**Kate Thompson, James Milligan, Angela Murphy, Ghazala Tabasam**⬤**, Mark I. Johnson**⬤*

Centre for Pain Research, School of Health, Portland Building, Leeds Beckett University, Leeds, United Kingdom

* m.johnson@leedsbeckett.ac.uk

**Data Availability Statement:** Data [i.e., interview transcripts] cannot be shared publicly because none of the interviewees consented to data sharing. Processed, qualitative data which demonstrates

## Abstract

'Unmasking Pain' is an innovative artist-led project to explore creative approaches for telling stories of life with persistent pain. The purpose of this qualitative study was to explore and describe the lived experience of taking part in 'Unmasking Pain' from the perspectives of people living with pain (workshop attendees), artists (workshop facilitators) and pain rehabilitation specialists. Informed by descriptive phenomenology, data was collected via semi-structured interviews with three artists (workshop facilitators), five workshop attendees (all living with persistent pain) and one pain rehabilitation specialist. Data was analysed using thematic analysis for descriptive phenomenology. Three themes emerged from the analysis: Theme 1 Space to Breathe, reflects the *context* of 'Unmasking Pain' such as the conditions, environment and setting that people experienced; Theme 2 Flexibility to Participate, reflects the experience of the *process* of 'Unmasking Pain' such as the actions or internal workings/ behaviours of the experience; and Theme 3 Possibility for Change, reflects the *cognitive or emotional* experience of 'Unmasking Pain'. There were nine sub-themes: 1(a) Dissolved hierarchy, 1(b) Human connection, and 1(c) Personally meaningful; 2(a) Go with the flow, 2 (b) Enjoyable experience, and 2(c) Curiosity to step into the unknown; and 3(a) Reforming pain, 3(b) Liberating in the moment (now), and 3(c) Gives you a boost (future). In conclusion, the uniqueness of 'Unmasking Pain' was to creatively facilitate curiosity and exploration of pain in a non-clinical, sociocultural, humanised context, to enable the possibility of a different pain experience. The findings from this qualitative study exploring the lived experience of taking part in 'Unmasking Pain' demonstrates the power and opportunity of the Arts sector and artist-led activities to support people with chronic pain to have a better pain experience.

## Introduction

In healthcare, chronic pain is defined as pain that has persisted or recurred for three months or longer, or beyond that expected for tissue to heal [1]. It is estimated to affect over 20% of

how themes were constructed (e.g., notes and documentation of research team data analysis meetings) are available from Institutional Data Access (the Research Data Repository), Leeds Beckett University (https://doi.org/10.25448/lbu.27187059.v1).

**Funding:** The author(s) received no specific funding for this work.

**Competing interests:** All authors declare that the research was conducted in the absence of any commercial or financial relationships that could be construed as a potential conflict of interest. MIJ wishes to declare that in the previous 5 years, his employer has received income for expert consultancy activities from GSK plc, TENSCare Ltd, and LifeCare Ltd. that lie outside of the submitted work. MIJ also declares book royalties from Oxford University Press. This does not alter our adherence to PLOS ONE policies on sharing data and materials.

people worldwide, creating a significant burden on the lives of individuals, and health and care systems [2]. Pain is a complex subjective experience; people living with chronic pain report struggling to articulate their pain experience to others, including health care professionals [3, 4]. Research findings demonstrate that single word adjectives, often found in pain assessment tools, are inadequate to capture the complexity of pain [5]. However the use of metaphorical pain language can be helpful [5, 6]. Articulating the pain experience to others is critical for people experiencing chronic pain to be able to move forward [4]. Understanding individuals' pain experience is an important area in which to focus pain research and is critical for person centred care [7].

Previously, we discussed how art and creativity can provide a more encompassing understanding of pain experience and its management and how the lens of art could be used to develop conceptual thinking beyond the biomedical paradigm of pain, shifting our understanding of pain towards a 'whole person' approach [8]. A whole person approach [9] aligns to key strategic drivers for health and care in England, with Personalised Care as a core value, integrating care between health, community, and voluntary sectors [10]. For instance, telling personal stories and articulating narratives about experiences of chronic pain may enable a sense of validation and serve as a useful therapeutic tool [11]. Pain narratives transformed by images and blogs on social media have made the pain experience 'visible', eliciting empathy and building communities [12]. Telling and validation of personal stories can aid health journeys of people experiencing chronic pain [13], and listening to others' narratives can give voice to the suffering of chronic pain, making people feel that they are not alone [14].

'Unmasking Pain' is an innovative pilot project, funded by Arts Council England, to explore creative approaches for telling stories of life with persistent pain [8]. The 'Unmasking Pain' project aimed to view pain through a lens of art and creativity by exploring and co-creating, through artistic activity and creation, shared understandings about what it means to live with pain and to tell individual pain stories. 'Unmasking Pain' piloted their concept in three different geographical areas of the north of England, with culturally-diverse populations from inner city neighbourhoods who were living with pain associated with a disability or long-term health conditions. This included people from Asian, Asian British, Black British, White British and Other ethnic groups drawn from (a) local Parkinson's and Fibromyalgia support groups, some with limited mobility, (b) an arts and social change charity, based in a disadvantaged inner-city neighbourhood and (c) Syrian and Kurdish refugees with experience of injury in war and/or painful long-term conditions whose management had been hampered by the emotional trauma of being a displaced person. The core of the project involved inviting people with persistent pain to take part in creative workshops, engaging in activities such as dance, art, puppetry and clay work, to co-create and facilitate ways to explore and to tell individual pain stories. People were invited to attend either one-off 'pop-up' workshops or a programme of six, 90-minute workshops. People with persistent pain associated with a variety of conditions (including fibromyalgia, Parkinson's disease, arthritis, and cancer) were eligible to take part if they were willing to share their story, listen to the stories of others and prepared to take part in the creative process, i.e. 'to try new things', and to be open to scientific analyses. A summary of the project, including development and delivery of workshops is available in supplementary material in Johnson et al. (2023) [8].

There were several evaluation activities using mixed-methods alongside the project, including various measurements of indicators of physical and mental health such as activity, sleep, and physiological measures of stress that will be reported elsewhere. In this manuscript we describe the qualitative evaluation. Informed by descriptive phenomenology, the purpose of this study was to explore and describe the lived experience of taking part in 'Unmasking Pain' creative workshops from the perspectives of people living with pain (workshop attendees),

artists (workshop facilitators) and pain rehabilitation specialists (involved in assisting the workshop facilitators). The objectives were to explore *what* and *how* people experienced the 'Unmasking Pain' workshops.

## Materials and methods

Ethical approval for this study was gained from Leeds Beckett University Research Ethics Committee. Application Ref: 102009. The study was advertised at 'Unmasking Pain' workshops and contact details of the research team were provided so that individuals could independently make contact should they want to take part. Individuals who made contact were provided with a participant information sheet. Informed consent (written and/or verbal) was taken from people who agreed to participate in the study.

### Study design

We used qualitative research methods informed by descriptive phenomenology [15, 16]. The focus of descriptive phenomenology is to explore and describe shared lived experiences [17, 18]. In keeping with descriptive phenomenology methodology, we did not use a theoretical framework to inform the interviews or the analysis [18]. We aimed to 'bracket' our disciplinary knowledge and own experiences via critical awareness and discussion, to mitigate the effects of preconceptions [19, 20], whilst acknowledging that complete objectivity is unattainable. Our analysis was guided by an established method of thematic analysis developed specifically for descriptive phenomenology [21]. We used Shorey and Ng's practical implications on the use of descriptive phenomenology methods [18] and COREQ guidance to guide our methodology and develop this manuscript [22].

### Recruitment of sample

Purposive sampling was used to select study participants with experience of the 'Unmasking Pain' workshops. This included people with persistent pain (greater than 3 months duration) who had taken part in the workshops (workshop attendees), artists (some who also live with persistent pain) who facilitated the workshops (workshop facilitators), and support staff that included pain rehabilitation specialists. The study invitation (study poster advert) was distributed by the organisers of 'Unmasking Pain' verbally and/or via email and/or via print to all people attending the 'Unmasking Pain' workshops.

The research team were keen to understand the views and experiences of these different groups of people, to explore the experience of the 'Unmasking Pain' workshops.

The recruitment period for this study started on 27 September 2022 and ended on 4 March 2023. The study was advertised at 'Unmasking Pain' workshops with contact details of the research team included, so that individuals could independently make contact should they want to take part. Individuals who made contact and wanted more information were provided with a participant information sheet. Informed consent was taken verbally and was recorded verbatim from people who agreed to participate in the study. All participants were offered the choice of providing either written or verbal consent; all participants chose verbal consent due to challenges in navigating technology and ease of practicalities (e.g. downloading/signing/returning information). All participants were adults, and no consent was waived.

### Data collection

Data was collected via semi-structured interviews which were conducted either in-person or via Microsoft Teams depending on participant preference and/or geographical location.

Interviews were conducted by two authors (KT and GT), and were audio recorded and transcribed verbatim. Interviewers took notes during the interviews to supplement the recordings. Interview transcripts were checked against recordings for accuracy by interviewers; they were not returned to participants. The interviewers had no relationship with interviewees prior to the study commencing, and interviewees were not provided with information about the interviewers, other than what was included in the study information sheet. We used Bevan [23] to inform the development of an interview guide for descriptive phenomenology. We piloted the interview guide with an individual who had attended one of the 'Unmasking Pain' workshops to test the questions for clarity and to explore if they facilitated the interviewee to talk about their experiences of 'Unmasking Pain' (interview date: 27 September 2022). Following the pilot, the ultimate interview guide was agreed by the research team (Supplementary file 1). The first interview took place on 4 October 2022 and the final interview on 3 March 2023. Interview transcripts were managed using NVivo [24].

## Data analysis

An established method of thematic analysis for descriptive phenomenology was used to guide our analysis [21]. To ensure consistency across the team we took the following approach:

**Achieve familiarity with the data through open-minded reading [21].** In the first stage of analysis each member of the research team (KT, GT, AM, JM, MIJ) independently familiarised themselves with the *same* interview transcript. The transcript was that of the first full interview participant (not the pilot interview). We each read this first transcript several times and highlighted (coded) sentences/phrases/paragraphs that answered the objectives of the study i.e. we highlighted participants descriptions of *what* was experienced at 'Unmasking Pain' and *how* it was experienced. We met in pairs, then as a full team, to discuss and compare our process after which we agreed a set of operational principles to inform how we would code all subsequent interview transcripts. These were:

i.  to read transcripts with an open mind, without a theoretical lens to guide the analysis

ii.  to pay attention to the words, perspectives, views and experiences described by participants, rather than reflecting on our own views and experiences

iii.  to ground the analysis in the words and experiences of participants and to attempt to mitigate our own biases

iv.  to use 'in vivo' (or as close to 'in vivo' codes as possible) to capture the essence of the sentences/phrases/paragraphs that describe *what* was experienced and *how* it was experienced [25]

**Search for meanings and themes [21].** In the second stage of analysis, the research team independently coded the rest of the interview transcripts (i.e. transcripts 2–9). We followed our agreed principles, systematically highlighting and allocating 'in vivo' codes to significant phrases/statements/sentences that captured *what* participants experienced and *how* it was experienced. NVivo was used to code all the transcripts. One author (KT) cross-examined all NVivo data that had been coded by the research team, to compare similarities and differences, and organise preliminary patterns as they emerged. The preliminary data patterns were reviewed by the full research team several times. We debated, analysed and discussed the patterns at three separate data meetings until we reached consensus and agreed the data patterns. These agreed patterns of data became the 'sub-themes' of our results.

**Organise themes into a meaningful whole** [21].   In our final data meeting, we checked that the names of the sub-themes accurately represented the verbatim quotes. We checked that the experience of 'Unmasking Pain', as described by participants, was not lost and that the names that we allocated to sub-themes reflected meaning from the perspective of participants. We discussed differences and commonality between the sub-themes, agreeing that the experience of 'Unmasking Pain' circled around three overarching themes. We kept recordings of our conversations, which were transcribed and kept as a data log. One author (KT) used the data log to produce a description of the three overarching themes, which were subsequently reviewed and revised by the full research team. This allowed the patterns to be genuinely co-created by the team, attempting to mitigate bias from one author.

## Results

Nine single in-depth semi-structured interviews were conducted with three workshop facilitators (artists), five workshop attendees and one pain rehabilitation specialist (7 female and 2 male participants). All workshop attendees lived with long term health conditions, often with co-morbidities and had experienced pain for many years. Individual interviews lasted on average 50 minutes (range 34 minutes to 69 minutes. Two further people contacted the authors but were unable to take part due to time constraints. There were no dropouts. All study participants are identified by participant number only in this manuscript.

Data saturation was initially identified by one author (KT) who had oversight of all data analysis which was managed using NVivo [24]. This author brought patterns of data to the wider team for discussion, who agreed that data saturation was evident after six interviews, i.e., when analysing data in the seventh interview transcript, all data was being sorted into existing nodes that had already evolved during analysis of the first six transcripts. A total of nine interview transcripts were analysed and included in the results. Data analysis revealed three themes and nine sub-themes (patterns of data) (Fig 1).

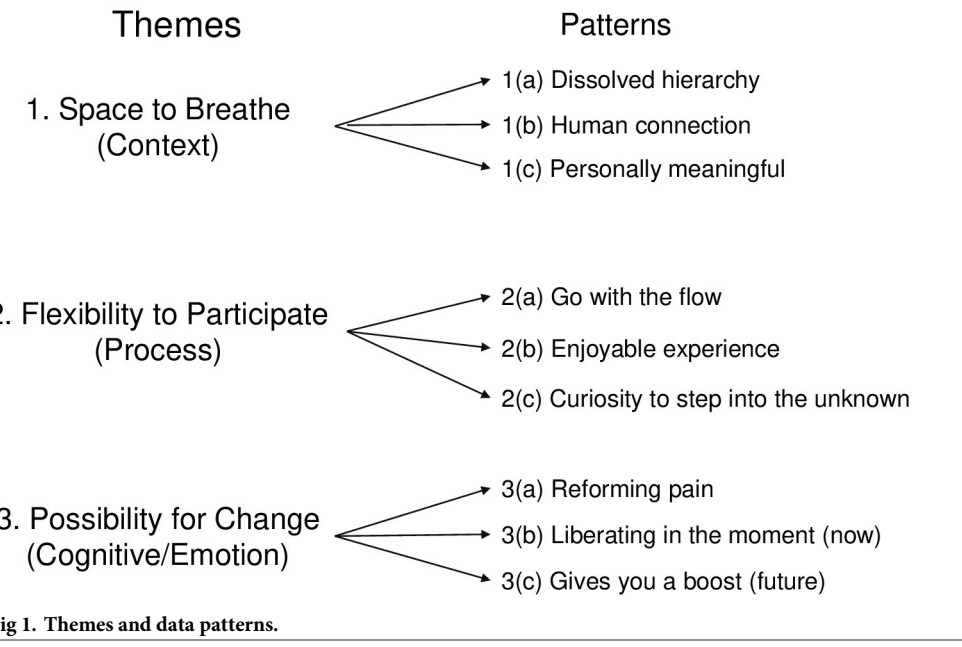

# The Experience of Unmasking Pain

**Fig 1. Themes and data patterns.**

Each of the three themes captures a different dimension of the experience of 'Unmasking Pain'. Theme 1 Space to Breathe: reflects the *context* of 'Unmasking Pain' i.e. the conditions, environment and setting that people experienced. Theme 2 Flexibility to Participate: reflects the experience of the *process* of 'Unmasking Pain' i.e. the actions or internal workings/behaviours of the experience. Theme 3 Possibility for Change: reflects the *cognitive or emotional* experience of 'Unmasking Pain'. Each theme incorporates the different views from study participants which included workshop attendees, workshop facilitators (artists) and a pain rehabilitation specialist. The verbatim quotes in the sub-themes demonstrate the different perspectives.

## Theme 1: Space to breathe

The theme '*Space to Breathe*' captures study participants descriptions of 'Unmasking Pain' as somewhere that workshop attendees, workshop facilitators (artists) and potentially pain rehabilitation specialists could be themselves, reflect and take stock of living with persistent pain. 'Unmasking Pain' was delivered in a non-clinical context where workshop attendees were put at ease by the workshop facilitators (artists). There are three patterns of data (sub-themes) within this theme which include (a) Dissolved hierarchy, (b) Human connection, and (c) Personally meaningful.

**Sub-theme 1(a) Dissolved hierarchy.** Dissolved hierarchy reflects study participants' descriptions of experiencing workshop attendees and workshop facilitators (artists) being on an equal footing, where activities were co-produced. There were no set or rigid rules. One workshop attendee described it as 'breaking down those barriers' that they had perceived in some previous healthcare settings.

*It became very much a two-way process. It was with the participants initially and then the bonding process moved out to the to the artist, what are they doing, why they're doing this, what we hope to achieve. And I think we were teaching them as much as they were teaching us [Participant 3–workshop attendee]*

*Some of the dancers actually have pain themselves and through interpretive dance, shows how they feel. And then they invite us to join in. Explain how we feel, how it relates to us. [Participant 9–workshop attendee]*

Workshop facilitators (artists) described a symbiotic experience, purposely facilitating workshops that allowed attendees and artists to co-produce creative stories together.

*I'm an artist, but I'm also clearly when I'm doing it, I'm clearly a participant. [Participant 4–workshop facilitator (artist)]*

*The participants are working with artists and the artists are talking about their pain as well. There's a symbiotic relationship [Participant 6–workshop facilitator (artist)]*

Workshop facilitators (artists) deliberately created an environment where workshop attendees might feel calm, and in the words of one of the workshop facilitators (artist) give workshop attendees 'space to breathe'.

*We do different things just to stimulate the feeling or give them the space [the participants], the space to feel that it's okay for them to have this pain and feel it and experience it. We're not trying to stop it, but giving them the space to breathe [Participant 5–artist]*

**Sub-theme 1(b) Human connection.** Human connection encapsulates study participants descriptions of a person-centred experience where social bonds and sharing with others was valued.

*The pain clinic was good, but it's very clinical. They don't give you the social side, so to speak. Where this was more - it didn't feel like they were teaching you things. It didn't feel rigid. [Participant 9–workshop attendee]*

Workshop attendees described that they felt supported; they trusted the workshop facilitators (artists) to lead and pace the activities.

*You wouldn't really want to be pulled or moved in a way that would make you painful. You know what I mean? They were very gentle and very supportive as in themselves. The actual physicality of it. Very gentle, very supportive. [Participant 2–workshop attendee]*

This was echoed by workshop facilitators (artists) in descriptions of the experience, who placed great value on compassion and consideration for others. The workshop facilitators (artists) wanted the workshop attendees to feel welcome and cared for, paying particular attention to the environment.

*[It was important for artists to] make those relationships and for the trust to be there [Participant 7 –Pain rehabilitation specialist]*

*There's art everywhere, and there's always flowing tea and stuff, and we had this really lovely big, big light room, which was amazing [Participant 6–workshop facilitator (artist)]*

**Sub-theme 1(c) Personally meaningful.** The data in this sub-theme was labelled 'personally meaningful' because it captures study participants' descriptions of an individual experience. For instance, workshop attendees experienced personalised attention to detail in the form of musical treatments, art and puppetry.

*I can't believe someone has just serenaded me with a flute. It was just such joy. It's so lovely. And it was so different, to think of your pain in that way as well. [Participant 2–workshop attendee]*

This deliberate attention to detail by the workshop facilitators (artists) was valued by workshop attendees who felt they were being treated as individuals, even in group situations. For some workshop attendees this resulted in feeling valued, cherished and cared for.

*We had the opportunity, to paint a mask, the colours that we thought of ourselves as being. And I must admit, I think I'm just all pastel colour. But I realized that that wasn't true, that I was pastel and red. I was a bit of an abstract [Participant 1–workshop attendee]*

*My Eureka moment came when I was introduced to my puppet. I was overwhelmed by the creation and the attention to detail. [Participant 3–workshop attendee]*

Workshop facilitators (artists) wanted to make the experience personally meaningful by getting to know the workshop attendees and tailoring creativity to them as individuals

*We asked them a lot of questions: what music do you like? What time of day do you like to listen to [music] and how do you think? And then they created musical treatments around that. But that idea that you have your own piece of music to listen to. That takes you off somewhere. I mean, genius. Brilliant. [Participant 6–workshop facilitator (artist)]*

### Theme 2: Flexibility to participate

This theme captures study participants descriptions of the experience of 'Unmasking Pain' whereby there was flexibility to participate without pressure or expectation. Participation was encouraged, but workshop attendees were able to participate on their own terms and were able to opt out without judgement. There are three data patterns (sub-themes) within this theme which include (a) Go with the flow, (b) Enjoyable Experience, and (c) Curiosity to step into the unknown.

**Sub-theme 2(a) Go with the flow.** At 'Unmasking Pain', workshop attendees were free to explore activities without being forced or overly persuaded. The level of participation was led by the workshop attendees.

*It was unscripted and it was how you felt in the moment. Yeah, and the flexibility to change direction too [Participant 3–workshop attendee]*

*There's an element of the artist just presenting something, so you know this is something I'd like to show you. Sort of gently, and from that there might be some involvement. You know, it's up to you what you do [Participant 7–Pain rehabilitation specialist]*

Workshop facilitators (artists) created a relaxed and inclusive environment. They were comfortable to take the workshops in different directions depending on how workshop attendees responded.

*There's flexibility not to participate in activity, to participate in an activity on your own terms, not what everybody else is doing [Participant 4–workshop facilitator (artist)]*

One workshop attendee kept a diary of their experience of 'Unmasking Pain', which they read out during the recorded interview.

*When I joined 'Unmasking Pain', I had no idea what it was about, where it would take me. I decided to go with the flow, relax, be spontaneous and let my instincts lead the way. So began an enthralling experience. Intriguing, imaginative and healing. [Participant 3–workshop attendee]*

**Sub-theme 2(b) Enjoyable experience.** The workshop facilitators (artists) purposefully circled around pain, putting enjoyable activities at the heart of the sessions.

*We just didn't mention pain. We just did enjoyable activities. I'm very, you know, great to be involved with, a real privilege to be involved with them [Participant 4–workshop facilitator (artist)]*

Workshop attendees described the experience as 'having a good time' and, to their mind, ironically 'having fun with pain'.

*We all really had a good time. It was like … a party every month. It was quite child-like in some ways, you know, you could unleash the inner child and play with paints and clay and just be as creative as you want it to be. It was having fun with pain, and that might sound completely ironic, but it was. I really enjoyed it, it was a unique experience. [Participant 3–workshop attendee]*

*It was fabulous, it really was. [Participant 9–workshop attendee]*

*I mean, I loved every minute. I was gutted when it finished 'cause, oh, not coming back for another four weeks. I really did thoroughly enjoy it. [Participant 8–workshop attendee]*

**Sub-theme 2(c) Curiosity to step into the unknown.**   This sub-theme captures the experience that 'Unmasking Pain' required both workshop facilitators (artists) and workshop attendees to navigate new and unfamiliar territory, to 'step into the unknown'. The process was unscripted and iterative with workshop facilitators (artists) and workshop attendees often unsure of what the outcome would be.

*We did all sorts of different things, it was like going into the unknown every time you went into that room. I mean, sometimes you look and think, what's that woman doing with a fishing rod? [Participant 3–workshop attendee]*

The research team discussed how this must have required an element of 'trust in the process', to explore and engage in this new experience. Hence, 'stepping in to the unknown'.

*I always look outside the box and I thought, well, this is going to be interesting and I'm either gonna love it or hate it. But I embraced it well and truly. [Participant 1–workshop attendee]*

*It opened my eyes to a lot of things. It really did. [Participant 8–workshop attendee]*

## Theme 3: Possibility for change

This theme captures patterns of data that relate to how the experience of 'Unmasking Pain' impacted how participants thought or felt about their pain. Study participants provided practical examples of how pain was explored creatively by expression of paint, clay work, puppetry dance and more. Patterns of data (sub-themes) within this theme include: (a) Reforming pain, (b) Liberating in the moment (now), and (c) Gives you a boost (future).

**Sub-theme 3(a) Re-forming pain.**   This pattern of data captures study participants descriptions of giving pain new identity or form, i.e. creatively giving pain shape, structure and appearance. For some workshop attendees, this is not something that they had previously had the opportunity to experience.

*It was amazing. It was absolutely amazing. I just sat there and looked at it [the claywork] and I honed it a bit and it was still really rough looking and it had knees and stuff like that and it was coming up out of the clay [Participant 2–workshop attendee]*

*[We created] a long wig sort of the thing, that looked like hair and we could plait it. And that was like the tightness in the muscles, are hard. You could tug on it to show everybody just how taut the muscles get in my body. [Participant 1–workshop attendee]*

From the perspective of workshop facilitators (artists), the purpose of facilitating workshop attendees to give pain form or identity was to explore it on their own terms, to give back some control.

*So, I guess by identifying as an external thing. Using the different things like a painting or a ball or clay or anything, we see it, externally. How you imagine it feels, how the pain feels like we're seeing it. In different forms. [Participant 5–workshop facilitators (artists)]*

*It sounds very sort of wibbly wobbly, but just to explore their pain with their hands, with the clay, whether they wanted to explore that. They produced an amazing range of work from really quite anatomical pieces, you know, with lungs and heart and knee and things like that and splints and internal pieces of metal that they put in using sticks and things like that … to totally abstract, expressive pieces that express something about what they're feeling about their pain [Participant 4–workshop facilitator (artist)]*

Viewing pain creatively, and articulating it as an external thing, allowed workshop attendees to explore and think about their pain in different ways.

*And you know having to think about what colour is pain. And what colour did I think was happiness? [Participant 1–workshop attendee]*

*I made a garden … I made ground into a garden with this Monster coming out of it. And it was supposed to represent our pain. That pain was just perfect, and I've never even thought about it before. [Participant 2–workshop attendee]*

**Sub-theme 3(b) Liberating in the moment (now).**   Study participants described that workshop attendees often felt liberated and distracted from pain in the moment, whilst enjoying the activities at the workshops.

*We did this fantastic drumming session. And we had silks on and they let us keep the sticks that we were given. And it was just fabulous and for that whole session I forgot I was in pain. [Participant 9–workshop attendee]*

*To think that the form of a dance can distract you so much that you don't have that immense pain. It's just so liberating. [Participant 6–workshop facilitator (artist)]*

The research team discussed how, in some instances, this could lead to feeling more empowered and in control, and less entrenched by the pain experience.

*People forgot about their pain. That's the most enjoyable thing to watch [Participant 1–workshop attendee]*

*So, we can practice [activities] it to help us with our pain. Now, when you've got your mind on something [activities] It helps to take your mind off your pain. [Participant 8–workshop attendee]*

**Sub-theme 3(c) Gives you a boost (future).**   In addition to feeling liberated from pain in the moment, workshop attendees described changes in their longer-term outlook, i.e. they found new ways to view pain. Workshop attendees described how they became open to the

prospect of alternatives and possibilities that they had not necessarily experienced before, or for a long time.

> *It's just like, opened up the world. You know, it opened up a different viewpoint [Participant 1–workshop attendee]*

> *It [the experience of 'Unmasking Pain'] gave me a boost actually to carry on and hope for the future really. [Participant 3–workshop attendee]*

Some workshop attendees described change that had already occurred, i.e. that they had made changes, others described carry over beyond the 'Unmasking Pain' workshops.

> *We talked about, how we can as a group or as an individual, help ourselves, to expand on what we can do to live a better life. That's what I took from it, it opened my eyes to different kinds of things that I can do. [Participant 8–workshop attendee]*

> *It's not something that I've done, and I can now say 'alright I've done that'. It's something that I've done, and I've continued to do because it helps me. [Participant 8–workshop attendee]*

## Discussion

The creative director of 'Unmasking Pain' set out to explore, through artistic activity and creation, shared understandings about what it means to live with pain [8]. This qualitative phenomenological study found that 'Unmasking Pain' facilitated a space to experience and explore pain with curiosity, in a non-clinical, sociocultural, humanised context; to enable the possibility of a different interpretation of pain experience.

Workshop attendees experienced a deep sense of empowerment and autonomy as they engaged with 'Unmasking Pain' on their own terms. The creative activities provided a safe and open space where attendees could explore and externalise their pain, giving it a unique identity that resonated with their personal experiences. Many participants described feeling a mix of curiosity and vulnerability as they ventured into the unknown, deconstructing their pain in ways they had never done before. The workshops were often described as transformative, not just enjoyable; the atmosphere was marked by a genuine sense of equality and human connection, which helped to dissolve any feelings of isolation or hierarchy. Participants felt a profound liberation from their pain, experiencing a freedom that they had not thought possible previously. These newfound perspectives and possibilities for the future, offered them hope and a renewed outlook on their journey with pain.

### 'Unmasking Pain' and arts interventions

The Arts have had a significant and important contribution to health and well-being over recent decades with the World Health Organization (WHO) wanting governments to include the arts in public health policy [26] based on a WHO scoping review of over 3000 studies that demonstrated the major role of the arts in health promotion, prevention and management [27]. Whilst the WHO review is not specifically about persistent pain, it demonstrates the positive impact of Arts in the management and treatment of many long-term health conditions, of which pain is a significant symptom. Specifically, Arts on Prescription, a social prescribing initiative in England, has shown benefits for people experiencing persistent pain. Dance-based intervention programs have been shown to be effective in reducing pain, depression, impact of

disease and anxiety, and increasing quality of life and physical function in people with fibromyalgia [28].

A scoping review by Fancourt and Finn [27] identified many health and well-being benefits of active co-creation of art in its many forms. Our phenomenological study identified multiple advantages of participating in 'Unmasking Pain'. This is emphasised in the overarching study themes, which highlight the often-unanticipated new perspectives that study participants with and without pain explored through engagement. Examples of this are shown in the data patterns that support the themes. Similarly, workshop attendees and workshop facilitators (artists) living with persistent pain described a humanising perspective of their pain experience (also highlighted in Fancourt and Saoirse's work) facilitated through an empowering and collaborative environment with diminished hierarchies. None of the workshop attendees described finding the challenging nature of participation difficult or threatening and instead described a safe space for exploration and making sense of their pain experience. They described appreciating the freedom to take-up the challenge of taking part while feeling confident they could opt out, which may have paradoxically added elements of safety to the explorative space.

There was evidence the workshop facilitators (artists) intentionally created an atmosphere of dissolved hierarchies, in part through sharing their own pain experiences. Study participants contrasted this experience to that of clinical settings which were highlighted as being reductionist in resources and perspectives. 'Unmasking Pain' appears to have facilitated changes in some attitudes and behaviours towards pain although the longevity of that cannot be ascertained from this study.

There are many examples of therapeutic experiences of art shown throughout the themes identified in this phenomenology study. For example, participants referred to an element of freedom resulting from the distracting nature of participation and engagement. This was shown to contribute to the transitional nature of taking part in workshop activities whereby the pre-existing identity and experience of pain was challenged, adding new perspectives which may have incorporated a less defining and dominating characterisation with transformative implications for the future. This is demonstrative of the powerful effects of art. Toye et al., [13] suggest that encouraging people with pain to connect with a meaningful sense of self and explore new possibilities for the future may influence healing journeys.

Koebner suggests two broad approaches that intersect art and pain. These are 1. Engaging with art without specifically discussing pain and 2. Using art to actively express pain or painful experiences [29]. Individuals with persistent pain who participated in exploring museums to address pain-related loneliness and social isolation reported relief of pain severity and unpleasantness and improvement in social connection [30], supporting our findings from this 'Unmasking Pain' study that indicates the importance of addressing social dimensions of pain experience. The uniqueness of 'Unmasking Pain' is that it was grounded, from conceptualisation to delivery, in the creative Arts discipline and outside of a health and care lens or setting. The creative director and artists are not from a health, care or therapy background, using pain rehabilitation specialists as a 'sounding board'. As 'Unmasking Pain' was conceived and led by artists, it did not have a 'medicalised' or 'therapy' viewpoint nor a 'de-medicalised' goal or endpoint. Nevertheless it appears to have had therapeutic effects.

## 'Unmasking Pain' and healthcare

Currently, in the United Kingdom, pathways of care for people with chronic pain are typically medical- or therapy-led clinics or pain management programmes, located in medical and/or healthcare settings [31, 32]. Whilst people with chronic pain are likely to have access to arts and creativity in local communities, via social prescribing networks, typically these do not

form part of core pain pathways. System change is required on a broader scale to facilitate seamless pathways to Arts interventions as part of the routine offer of a pain pathway of care in the health service. An example of one such approach is Rethinking Pain, a community-based pain support service in Bradford District and Craven in England led by Keighley Healthy Living, a voluntary and community sector charity [33, 34].

If 'Unmasking Pain' is viewed through a health and care lens, it broadly sits within the field of social prescribing or 'social interventions' for people with chronic pain. Social prescribing for other health conditions has seen significant and positive impacts including increased self-esteem and confidence; improvement in mental well-being and positive mood; and reduction in anxiety, depression and negative mood [35]. An example of social prescribing is 'Arts on prescription', where arts are 'prescribed' to individuals by healthcare professionals and/or social prescribers to improve their health and wellbeing. Arts on prescription originated to support the mental and/or physical wellbeing of individuals, rather than specifically for people with chronic pain, though the concepts are portable. Arts on prescription is distinct from 'art therapy' because the focus is on activity and social interaction, rather than the person's medical condition [36]. The main strength of social prescribing is to connect clinical, community and voluntary services and support. Challenges include sourcing appropriate resources, equity and access (particularly for marginalised or disadvantaged populations), managing appropriate referrals, health responsibility, lack of communication between providers, and unknown long term impacts [37, 38]. These are important considerations if interventions like 'Unmasking Pain' are to be scaled up as part of health and care pathways.

## Strengths and weaknesses of the study

The authorship team have diverse multidisciplinary experience with professional backgrounds in physiotherapy, occupational therapy, biomedical sciences, physiology, and pain science. The authors have experience working across a range of environments with particular focus on pain, qualitative research, health, and education. As a result, the research team brought diverse expertise from various fields, enhancing the depth and breadth of the study. Integrating different perspectives enabled creative thinking with each team member contributing their unique insights and skills.

In regards to rigour, Lincoln and Guba describe four essential criteria (credibility, transferability, dependability and confirmability) for qualitative research [39, 40]. We aimed to ensure that research findings described in this manuscript are *credible* by including sufficiently meaningful verbatim quotes as supporting evidence. We address *dependability* by providing detail of our processes, so that the study could be repeated. We took steps to address *confirmability* by carefully considering and checking our data analysis reflected participants words and experiences, openly discussing our biases and checking final themes against the data in the interview transcripts. We provide detailed information about the context of 'Unmasking Pain', so that readers can decide if *transferability* is possible.

Furthermore, we ensured rigour by using an interview guide specifically developed for descriptive phenomenology. This provided consistency between interviews but also allowed participants to openly describe their experiences with interviewer bias minimised. Interviews were transcribed verbatim, and all text was 'coded' where possible using in-vivo codes and participants words. Following pilot data analysis, we aimed for consistency by one author (KT) acting as a second person to review the research team's data analysis. We used an established method of thematic data analysis for descriptive phenomenology. The full research team met at regular intervals to debrief and cross examine patterns and themes as they evolved. When constructing patterns (KT) kept memos, reflecting on biases and factors that might prejudice

the analysis. For example, as academics, researchers and clinicians we recognised and noted that our personal background, experiences, and assumptions may shape the interpretation of the data. For instance, KT has previously worked in clinical pain services which may have resulted in more focus on contrasting environmental factors. To overcome this, throughout data analysis meetings the authorship team discussed their experiences and positionality as researchers. By keeping a conscious awareness of these factors, the team hoped to mitigate bias and maintain focus on giving an authentic voice to the perspectives of participants. Nevertheless, we acknowledge that complete objectivity is unattainable. Bracketing of the preconceptions was managed through critical awareness and discussion but we recognise that all qualitative research depends on the subjectivities of the researchers as they make decisions and judgements relating to the content and importance of data. While best efforts to manage preconceptions were made, we could have enhanced this process with journaling or research diaries to further mitigate our effects on the research process.

## Conclusion

In conclusion, this descriptive phenomenological study demonstrates the uniqueness of 'Unmasking Pain' to creatively facilitate curiosity and exploration of pain in a non-clinical, sociocultural, humanised context, to enable the possibility of a different pain experience. Furthermore, the findings demonstrate the power and opportunity of the Arts sector and artist-led activities to support people with chronic pain to have a better pain experience.

## Supporting information

**S1 File. Interview guide and COREQ checklist.**
(DOCX)

## Acknowledgments

The following people are acknowledged for the design and/or delivery and/or administrative support of the 'Unmasking Pain' project: Balbir Singh, Paul Chazot, Frances Cole, Rosemary Cruickshank, Dawn Fuller, Charlotte Keyse, Emma Tregidden, James Woodall, Kali Chandrasegaram, Devika Rao, Vilmore James, Adam Strickson, Louise Grassby, Sam White, Aniruddha Mukherjee, Jenny Wilson, Ford Collier, Sarah Mason, Natasha Joseph, Sarah Partridge, Chris Speyer, Madhura Karandikar, Mansi Dabral, Malcolm Johnson, Tim Smith, Karol Wyszynski, Paul Floyd Blake, Julian Germain, Gareth Dakin, Nathan Towers, Mark Baker, Elia Tome, Susan Burns, Holly Amos, Ezekiel Oliveira, Jess Williams, Tammy Tsang, Tim Ingram, David Andrassy, Jordan Mereil, Anamaria Wills, Bisakha Sarker, Anna Harpin, and Bobak Champion.

## Author Contributions

**Conceptualization:** Kate Thompson, Angela Murphy, Ghazala Tabasam, Mark I. Johnson.

**Data curation:** Kate Thompson.

**Formal analysis:** Kate Thompson, James Milligan, Angela Murphy, Ghazala Tabasam, Mark I. Johnson.

**Investigation:** Kate Thompson, James Milligan, Angela Murphy, Ghazala Tabasam, Mark I. Johnson.

**Methodology:** Kate Thompson, James Milligan, Angela Murphy, Ghazala Tabasam, Mark I. Johnson.

**Project administration:** Kate Thompson.

**Resources:** Kate Thompson, Mark I. Johnson.

**Software:** Kate Thompson.

**Supervision:** Kate Thompson, Mark I. Johnson.

**Validation:** Kate Thompson, James Milligan, Angela Murphy, Ghazala Tabasam, Mark I. Johnson.

**Visualization:** Kate Thompson, James Milligan, Angela Murphy, Ghazala Tabasam, Mark I. Johnson.

**Writing – original draft:** Kate Thompson.

**Writing – review & editing:** Kate Thompson, James Milligan, Angela Murphy, Ghazala Tabasam, Mark I. Johnson.

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
