## [Decision Letter · Decision Letter 0]

29 Aug 2024

PONE-D-24-11622Unmasking pain through creativity: A phenomenological study of person-centred pain exploration for people living with chronic painPLOS ONE

Dear Dr. Johnson,

Thank you for submitting your manuscript to PLOS ONE. After careful consideration, we feel that it has merit but does not fully meet PLOS ONE’s publication criteria as it currently stands. Therefore, we invite you to submit a revised version of the manuscript that addresses the points raised during the review process.

We look forward to receiving your revised manuscript.

Kind regards,

Stefaan Six, Ph.D.

Academic Editor

PLOS ONE

“All authors declare that the research was conducted in the absence of any commercial or financial relationships that could be construed as a potential conflict of interest. MIJ wishes to declare that in the previous 5 years, his employer has received income for expert consultancy activities from GlaxoSmithKline, TENSCare, and LifeCare Ltd. that lie outside of the submitted work. MIJ also declares book royalties from Oxford University Press.”

Reviewers' comments:

Reviewer's Responses to Questions

**Comments to the Author**

1. Is the manuscript technically sound, and do the data support the conclusions?

Reviewer #1: Yes

Reviewer #2: Yes

2. Has the statistical analysis been performed appropriately and rigorously? 

Reviewer #1: Yes

Reviewer #2: N/A

3. Have the authors made all data underlying the findings in their manuscript fully available?

Reviewer #1: No

Reviewer #2: Yes

4. Is the manuscript presented in an intelligible fashion and written in standard English?

Reviewer #1: Yes

Reviewer #2: Yes

5. Review Comments to the Author

Reviewer #1: I commend the authors for undertaking this work and points raised are to be clarified in the manuscript.

Line 2: Although three months is a common time used to define chronic pain, the authors should consider including literature that define chronic pain in relation to the presence of pain beyond tissue healing as well, and not just three months.

Line 23: Should be italised (or inverted commas) throughout the document to clearly indicate a name of the project and differentiate it from the rest of the sentence.

Line 28: "diverse populations such as..." The authors to name the populations.

Line 31: "once-off" vs one-off

Line 74: The authors should specify the main questions in this manuscript. Instead, going to the article by Bevan (2014) cited here, there is no interview guide in that article. Rather cite the supplementary file here instead of Bevan (2014).

Line 87-88: It is unclear why the authors specify "not the pilot interview" when line 80 clearly states that data from the pilot interview were not included in analysis.

Line 119-120: The authors have gone into detail to describe the analysis process to reduce the chances of bias, however, there is no mention of the rigor and other aspects of trustworthiness for this study. What are the reflexivity points of the authors? The reader is forced to take the findings at face value (beyond the discussions the authors had), and more is needed.

Line 126 (Table 1): The participants need a demographic table that includes more than what they do (artist/attendee). The opening line of this manuscript defined chronic pain >3 months, yet, we do not even know how long each participant has had the pain for, nor do we know long the attendees were part of the project. The sampling was purposive and simply having artist/attendee to describe the participants does not do justice to this paper.

It is not even about confidentiality, as the information mentioned above will not identify participants.

Line 130: The authors must specify how data saturation was identified. Was a saturation matrix used? etc.

Line 137-144: Throughout the manuscript, the authors cannot refer to the themes as Theme [1]/[2]/[3] when the reference system uses the same square brackets.

Line 147: the authors specify the workshop attendees and the artists, so the reader should assume that not the rehab specialist? Specify this in the manuscript.

Line 149: The authors should rather refer to the artists who were facilitating the workshops as "the workshop facilitators". This was, the reader will only associate artists with the ones that participated in this study (participants 4-6).

The discussion section does not adequately synthesise the enjoyment of the project's approach, how liberating it is and how one can have hope for the future as the participants expressed. It would have been better to have the discussion brought to a more qualitative feel, by staying close to the emotions and "feels" that the participants expressed while integrating with the literature.

Line 336-337: add a reference for this aim set out by the creative director.

Line 348-349: needs a reference

Line 403: a link should not be used as a reference in-text.

Line 420-439: while these pargraphs address my previous comment on the rigor of the study, the authors' reflexivity is not included, and biases from the authors' beliefs/assumptions and how they could have (or not) prejudiced the study is not touched on.

Reviewer #2: Thank you for this exemplary phenomenological work.

Materials and methods:

Study design

p. 5, lines 55-56: I suggest adding the completed COREQ checklist as a supplement.

Recruitment of sample

p. 4, lines 59-60: What was the recruitment rate for the study and what were the reasons for non-participation?

p. 5, lines 68-69: Why was informed consent given verbally rather than in writing?

Data collection

p. 5, lines 73-74: On what basis was the interview guide developed/grounded/were the interview guide questions based?

Results

p.6, lines 124-125: Could you please add the average interview duration?

Table 1: The table contains very little information. I suggest deleting the table and putting the information in the body text.

6. PLOS authors have the option to publish the peer review history of their article (what does this mean?). If published, this will include your full peer review and any attached files.

Reviewer #1: No

Reviewer #2: No

---

## [Author Response · Author response to Decision Letter 0]

11 Sep 2024

Responses to Reviewers' comments: (Also available in Cover Letter and in Response to Reviewers files). Line numbers relate to the Revised Tack Change manuscript, There has been some minor proof editing in the 'clean' mansucript. 

Comments to the Author

1. Is the manuscript technically sound, and do the data support the conclusions?

Reviewer #1: Yes

Reviewer #2: Yes

Authors’ Response: N/A

2. Has the statistical analysis been performed appropriately and rigorously? 

Reviewer #1: Yes

Reviewer #2: N/A

Authors’ Response: N/A

3. Have the authors made all data underlying the findings in their manuscript fully available?

Reviewer #1: No

Reviewer #2: Yes

Authors’ Response: Please see response to Editor in Cover Letter.

4. Is the manuscript presented in an intelligible fashion and written in standard English?

Reviewer #1: Yes

Reviewer #2: Yes

Authors’ Response: N/A

Response to Reviewers

5. Review Comments to the Author

Reviewer #1: I commend the authors for undertaking this work and points raised are to be clarified in the manuscript.

Line 2: Although three months is a common time used to define chronic pain, the authors should consider including literature that define chronic pain in relation to the presence of pain beyond tissue healing as well, and not just three months.

Authors’ Response: Amended 

Line 23: Should be italised (or inverted commas) throughout the document to clearly indicate a name of the project and differentiate it from the rest of the sentence.

Authors’ Response: Amended by inserting inverted commas throughout 

Line 28: "diverse populations such as..." The authors to name the populations.

Authors’ Response: Added detail, line 28 onwards. 

Line 31: "once-off" vs one-off

Authors’ Response: Amended

Line 74: The authors should specify the main questions in this manuscript. Instead, going to the article by Bevan (2014) cited here, there is no interview guide in that article. Rather cite the supplementary file here instead of Bevan (2014).

Authors’ Response: We have amended the text in the section Data collection but would still like to refer to Bevan to elaborate on questionnaire development in response to a comment from reviewer 2 

(line 102 onwards). 

Line 87-88: It is unclear why the authors specify "not the pilot interview" when line 80 clearly states that data from the pilot interview were not included in analysis.

Authors’ Response: Offending sentence deleted 

Line 119-120: The authors have gone into detail to describe the analysis process to reduce the chances of bias, however, there is no mention of the rigor and other aspects of trustworthiness for this study. What are the reflexivity points of the authors? The reader is forced to take the findings at face value (beyond the discussions the authors had), and more is needed.

Authors’ Response: Thank you for highlighting this important point. We have amended our text to directly address rigour and reflexivity (see section Strengths and weaknesses of the study and line 500 onwards). 

Line 126 (Table 1): The participants need a demographic table that includes more than what they do (artist/attendee). The opening line of this manuscript defined chronic pain >3 months, yet, we do not even know how long each participant has had the pain for, nor do we know long the attendees were part of the project. The sampling was purposive and simply having artist/attendee to describe the participants does not do justice to this paper. It is not even about confidentiality, as the information mentioned above will not identify participants.

Authors’ Response: We have removed Table 1 in line with the comment by reviewer 2. We have amended the text in the Results to provide more information about the characteristics of the study sample.

Line 130: The authors must specify how data saturation was identified. Was a saturation matrix used? etc.

Authors’ Response: We have amended Results section to clarify (line 175)

Line 137-144: Throughout the manuscript, the authors cannot refer to the themes as Theme [1]/[2]/[3] when the reference system uses the same square brackets.

Authors’ Response: Amended by removing brackets from themes and adding curly brackets for sub-themes 

Line 147: the authors specify the workshop attendees and the artists, so the reader should assume that not the rehab specialist? Specify this in the manuscript.

Authors’ Response: We have amended the text to clarify. 

Line 149: The authors should rather refer to the artists who were facilitating the workshops as "the workshop facilitators". This was, the reader will only associate artists with the ones that participated in this study (participants 4-6).

Authors’ Response: Thank you - we have amended the text in line with your suggestion.

The discussion section does not adequately synthesise the enjoyment of the project's approach, how liberating it is and how one can have hope for the future as the participants expressed. It would have been better to have the discussion brought to a more qualitative feel, by staying close to the emotions and "feels" that the participants expressed while integrating with the literature.

Authors’ Response: We have elaborated further on this point in the Discussion (line 424 onwards) 

Line 336-337: add a reference for this aim set out by the creative director.

Authors’ Response: Added

Line 348-349: needs a reference

Authors’ Response: Added

Line 403: a link should not be used as a reference in-text.

Authors’ Response: We have converted into an in-text reference

Line 420-439: while these pargraphs address my previous comment on the rigor of the study, the authors' reflexivity is not included, and biases from the authors' beliefs/assumptions and how they could have (or not) prejudiced the study is not touched on. 

Authors’ Response: We have added additional information regarding reflexivity into the Strengths and weaknesses of the study subsection (line 516 onwards).

Reviewer #2: Thank you for this exemplary phenomenological work.

Materials and methods:

Study design

p. 5, lines 55-56: I suggest adding the completed COREQ checklist as a supplement.

Recruitment of sample

Authors’ Response: We have provided a completed COREQ checklist and added it to supplementary file 1

p. 4, lines 59-60: What was the recruitment rate for the study and what were the reasons for non-participation?

Authors’ Response: We did not track ’recruitment rate’. The study invitation was distributed verbally and/or via email and/or via a study advert to all people who attended ‘Unmasking Pain’ workshops. Twelve people contacted the PI of which ten agreed to participate. Anecdotally, workshop facilitators reported to us that reasons people did not wish to participation was ‘time constraints’. We have clarified this by amending the sections Recruitment of sample and Results (~line 80 and 157 respectively).

p. 5, lines 68-69: Why was informed consent given verbally rather than in writing? 

Authors’ Response: All participants were sent written copies of the participant information sheet and consent form. All participants were offered the choice of providing either written or verbal consent; all participants chose verbal consent due to challenges in navigating technology and ease of practicalities (e.g. downloading/signing/returning information). We have added a sentence in the section Recruitment of sample to reflect this (~line 90). 

Data collection

p. 5, lines 73-74: On what basis was the interview guide developed/grounded/were the interview guide questions based?

Authors’ Response: We have amended the text to clarify this point and have elaborated on questionnaire development in response to a comment from reviewer 1 (~line 80 onwards).

Results

p.6, lines 124-125: Could you please add the average interview duration?

Authors’ Response: This has been added to the Results section (line 147)

Table 1: The table contains very little information. I suggest deleting the table and putting the information in the body text.

Authors’ Response: We have deleted the table and added further detail to the Results

---

## [Decision Letter · Decision Letter 1]

30 Sep 2024

Unmasking pain through creativity: A phenomenological study of person-centred pain exploration for people living with chronic pain

PONE-D-24-11622R1

Dear Dr. Johnson,

We’re pleased to inform you that your manuscript has been judged scientifically suitable for publication and will be formally accepted for publication once it meets all outstanding technical requirements.

Kind regards,

Stefaan Six, Ph.D.

Academic Editor

PLOS ONE

Additional Editor Comments (optional):

Reviewers' comments:

Reviewer's Responses to Questions

**Comments to the Author**

1. If the authors have adequately addressed your comments raised in a previous round of review and you feel that this manuscript is now acceptable for publication, you may indicate that here to bypass the “Comments to the Author” section, enter your conflict of interest statement in the “Confidential to Editor” section, and submit your "Accept" recommendation.

Reviewer #2: All comments have been addressed

2. Is the manuscript technically sound, and do the data support the conclusions?

Reviewer #2: Yes

3. Has the statistical analysis been performed appropriately and rigorously? 

Reviewer #2: N/A

4. Have the authors made all data underlying the findings in their manuscript fully available?

Reviewer #2: Yes

5. Is the manuscript presented in an intelligible fashion and written in standard English?

Reviewer #2: Yes

6. Review Comments to the Author

Reviewer #2: Thank you for your revision. All comments have been addressed. I recommend the acceptance of the manuscript.

7. PLOS authors have the option to publish the peer review history of their article (what does this mean?). If published, this will include your full peer review and any attached files.

Reviewer #2: No

---

## [Editor Report · Acceptance letter]

21 Oct 2024

PONE-D-24-11622R1 

PLOS ONE

Dear Dr. Johnson, 

I'm pleased to inform you that your manuscript has been deemed suitable for publication in PLOS ONE. Congratulations! Your manuscript is now being handed over to our production team.

Kind regards, 

on behalf of

Dr. Stefaan Six 

Academic Editor

PLOS ONE